# Analysis of Salivary Glands and Saliva from *Aedes albopictus* and *Aedes aegypti* Infected with Chikungunya Viruses

**DOI:** 10.3390/insects10020039

**Published:** 2019-02-01

**Authors:** Irma Sanchez-Vargas, Laura C. Harrington, William C. Black, Ken E. Olson

**Affiliations:** 1Arthropod-borne and Infectious Diseases Laboratory, Colorado State University, Ft. Collins, CO 80523, USA; Irma.Sanchez-Vargas@colostate.edu (I.S.-V.); william.black@colostate.edu (W.C.B.IV); 2Department of Entomology, Cornell University, Ithaca, NY 14850, USA; lch27@cornell.edu

**Keywords:** chikungunya virus, arbovirus, *Aedes albopictus*, *Aedes aegypti*, salivary glands, and saliva, salivary gland exit barrier

## Abstract

Chikungunya virus (CHIKV) is a medically important mosquito-borne virus transmitted to humans by infected *Aedes* (*Stegomyia*) species. In 2013–2014, *Ae. aegypti* transmitted CHIKV to humans in the Caribbean and in 2005–2006, *Ae. albopictus* transmitted CHIKV on La Réunion Island (Indian Ocean basin). CHIKV LR2006 OPY1 from the La Réunion epidemic was associated with a mutation (E1:A226V) in the viral E1 glycoprotein that enhanced CHIKV transmission by *Ae. albopictus*. CHIKV R99659 from the Caribbean outbreak did not have the E1:A226V mutation. Here, we analyzed the salivary glands and saliva of *Ae. albopictus* strains from New Jersey, Florida, Louisiana and La Réunion after infection with each virus to determine their transmission potential. We infected the *Ae. albopictus* strains with blood meals containing 3–7 × 10^7^ PFU/mL of each virus and analyzed the mosquitoes nine days later to maximize infection of their salivary glands. All four *Ae. albopictus* strains were highly susceptible to LR2006 OPY1 and R99659 viruses and their CHIKV disseminated infection rates (DIR) were statistically similar (*p* = 0.3916). The transmission efficiency rate (TER) was significantly lower for R99659 virus compared to LR2006 OPY1 virus in all *Ae. albopictus* strains and *Ae. aegypti* (Poza Rica) (*p* = 0.012) suggesting a salivary gland exit barrier to R99659 virus not seen with LR2006 OPY1 infections. If introduced, LR2006 OPY1 virus poses an increased risk of transmission by both *Aedes* species in the western hemisphere.

## 1. Introduction

*Chikungunya virus* (CHIKV; *Alphavirus*; *Togaviridae*) is an arthropod-borne virus (arbovirus) maintained during epidemics in transmission cycles between viremic humans and infected *Aedes* (*Stegomyia*) mosquitoes. The genome of CHIKV is a positive single stranded RNA (~11,800 nucleotides) containing two open reading frames (ORFs) between the 5′ and 3′ untranslated regions [1]. The first ORF, encodes the nonstructural polyprotein (nsP1-nsP4), and the second ORF encodes the structural proteins (C, E3, E2, 6K and E1). The two ORFS are separated by a small untranslated junction region [1]. CHIK disease in symptomatic patients presents as fever and intense rash, weakness, myalgia and polyarthralgia [2]. The onset of fever coincides with viremia, and the viral load can rapidly reach up to 10^9^ viral genome copies per milliliter of blood or virus titers of 10^7–9^ plaque-forming unit equivalents /mL [2,3]. The principle vector of CHIKV is *Ae. aegypti*, a peridomestic, day-biting mosquito found in tropical and subtropical regions of the world [2]. However, in 2005–2006 an outbreak of CHIKV occurred on La Réunion Island (Indian Ocean basin) that was transmitted by another day biting mosquito, *Ae. albopictus* [4]. Significantly, *Ae. albopictus* has invaded new regions globally. They now thrive in temperate areas of the western hemisphere and could pose additional health risks to a CHIKV susceptible human population [5]. 

CHIKV are classified in three genotypic groups: East-Central-South Africa (ECSA), West Africa, and Asian. After the 2006 La Réunion outbreak, a new Indian Ocean lineage (IOL) was identified as a monophyletic descendant of the ECSA group [6,7,8]. The RNA genomes of ECSA-IOL CHIKV were associated with a genetic mutation in the CHIKV E1 glycoprotein-coding region resulting in a single alanine to valine amino acid change (E1-A226V) that adapted the virus to *Ae. albopictus* [9,10,11,12]. Others have reported that ECSA-IOL CHIKV infection of *Ae. albopictus* virus can be further enhanced by amino acid changes in their E2 glycoprotein [13]. In 2013–2014, CHIKV (Asian genotype) spread to more than 22 countries in the western hemisphere and caused tens of thousands of human cases [14,15,16]. The R99659 CHIKV was isolated from the British Virgin Islands during an epidemic but CHIKVs of the ECSA-IOL genotype were not associated with this outbreak [15]. However, the ECSA-IOL CHIKV could rapidly spread in the western hemisphere if viremic individuals with the ECSA-IOL CHIKV genotype travel to the United States (U.S.) and infect established *Ae. albopictus* populations. Autochthonous transmission of ECSA-IOL CHIKV by *Ae. albopictus* could then lead to new outbreaks in CHIKV-naïve human populations [17].

Our goal was to analyze salivary gland and the titer of virus in saliva of the two CHIKV isolates, LR2006 OPY1 (ECSA-IOL genotype) and R99659 (Asian genotype), in strains of *Ae. albopictus* from New Jersey, Florida and Louisiana. We compared infections of these mosquitoes with an *Ae. albopictus* strain from La Réunion Island and an *Ae. aegypti* strain from Mexico (Poza Rica). The extrinsic incubation period (EIP), the time from initial acquisition of CHIKV by the mosquito until virus transmission ranges from 2 to 9 days [18]. We analyzed salivary gland and saliva infection status for CHIKV in each infected mosquito at 9 days post-blood meal infection. Typically, CHIKV infects the midgut when the mosquito imbibes a viremic blood meal. The virus then disseminates to other tissues including salivary glands where the virus enters saliva. Virus in saliva is transmitted to a susceptible host during acquisition of a subsequent blood meal. To evaluate dissemination and transmission rates, saliva was collected to determine transmission potential, and then salivary glands were dissected from the same mosquitoes to analyze virus dissemination. Here, we report dissemination rate (DIR) as the percent of total mosquitoes exposed to virus that were positive for virus in the salivary gland and the transmission efficiency rate (TER) as the percent of mosquitoes positive for virus in the saliva that were also positive for virus in the salivary glands. Vector competence was determined here by multiplying DIR × TER. We clearly showed under laboratory conditions that the Louisiana (Lake Charles), Florida, and New Jersey strains of *Ae. albopictus* had high DIR values. The TER values for CHIKVs were lower than expected for R99659 virus given the mosquitoes’ high DIRs and suggested the presence of a salivary gland escape barrier (SGEB) for that virus. All mosquitoes were competent to transmit LR2006 OPY1. Of the three U.S. *Ae. albopictus* strains, the Louisiana and Florida strains could serve as excellent vectors for CHIKV R99569 (Asian) transmission. The New Jersey *Ae. albopictus* strain had the lowest vector competence (DIR × TER) of the three *Ae. albopictus* strains for R99569 virus but had statistically similar vector competence as the other strains for LR2006 OPY1 (ECSA-IOL) [19]. 

## 2. Materials and Methods 

### 2.1. Virus and Cell Culture

Vero cells, LLC-MK2 monkey kidney cells and C6/36 (*A. albopictus*) cells were cultured in modified Eagle’s medium (MEM) supplemented with 8% fetal bovine serum, l-glutamine, non-essential amino acids and penicillin/streptomycin and maintained at 5% CO_2_ at 37 °C and 28 °C, respectively. The two low passage CHIKV isolates (R99659, British Virgin Islands, Asian genotype; Accession # KJ451624.1) and LR2006 OPY1 (La Réunion, ECSA genotype; Accession # DQ443544.2) were provided by the Centers for Diseases Control and Prevention (CDC-Fort Collins, CO, USA). The presence of the E1-A226V mutation in LR2006 OPY1 virus was confirmed by RT-PCR using specific primer sets designed to amplify the nucleotide sequence surrounding the mutation as previously described [20]. We infected Vero cells with low passage CHIKV at a 0.001 multiplicity of infection (MOI) then incubated cells at 37 °C for 72 h. Virus aliquots were stored at −80 °C until used for preparing infectious blood meals. 

### 2.2. Mosquitoes

*Aedes albopictus* from Florida, Louisiana (Lake Charles), and La Réunion Island (Indian Ocean) were provided by the Centers for Diseases Control and Prevention (CDC-Fort Collins, CO, USA). The New Jersey *Ae. albopictus* strain was obtained from Cornell University and the *Ae. aegypti* line was collected in Poza Rica, Mexico [4,21]. The La Réunion, New Jersey and Poza Rica mosquitoes were less than 10 generations in colony, Florida and Louisiana strains were more generations in colony. Mosquito eggs were hatched to obtain the immature stages and eclosed adults were held in cages at 28 ± 1 °C and 70–80% relative humidity. Adults were fed 10% sucrose ad libitum and maintained at 16:8 light:dark photoperiod. Groups of 200 adult females (1 week post-eclosion) were placed in 2.5 L cartons, deprived of sugar and water overnight to promote feeding of artificial blood meals consisting of virus-infected Vero cell and medium suspension (60% vol/vol), 40% (vol/vol) defibrinated sheep blood (Colorado Serum Co., Boulder, CO, USA) and 1 mM ATP [22]. Blood engorged mosquitoes were maintained at the indicated insectary conditions for 9 days. Virus titers (pfu/mL) in the blood meal were quantified using a plaque assay described below. 

### 2.3. Infectious Virus Titration by Plaque Assay

Plaque assays were performed using confluent monolayers of Vero cells in 24-well plates to determine virus titers in saliva and salivary glands. The samples were sterilized by passing them through Acrodisc HT Tuffryn 0.2-μm syringe filters (Pall Life Sciences, East Hills, NY, USA). Vero cells were infected for 1 h with 10-fold serial dilutions of saliva or salivary gland homogenate samples. Infected cells were overlaid with a 1% agar-nutrient mixture (agar solution (1g/77 mL DI H_2_O): nutrient solution (10 mL of 10×/100 mL Media 199, 7% heat inactivated fetal bovine serum (FBS), 7.5% sodium bicarbonate (4mL/100mL), 2% DEAE- dextrose in Hanks balanced solution (1mL/100mL), 0.5 mL/100mL MEM essential amino acids (15× solution), and 0.5 mL/100mL MEM vitamins (100× solution)). After 7 days incubation at 37 °C cells were stained with 3 mg/mL MTT (3-(4,5-dimethylthiazol-2-yl)-2,5-diphenyltetrazolium bromide) solution and incubated for 4 h [23,24]. Viral titers were determined by counting visible plaques and individual saliva or salivary gland titers were reported as pfu/mL. 

### 2.4. Saliva Collection and Salivary Gland Dissections

At day 9 post-infections, saliva from each female was collected using a previously described saliva collection method [25]. Briefly, females were chilled, and their wings and legs removed. The proboscis of the mosquito was inserted into a 1.0 µL micropipette (microcaps, Drummond Scientific Company, Broomall, PA, USA) filled with immersion oil type B and allowed to salivate into the oil at room temperature. After 30–45 min, the proboscis was removed from the capillary and oil containing the saliva was expelled under pressure into 1.5 mL tubes containing 300 μL DMEM medium (20% of heat inactivated FBS, 1% penicillin/streptomycin, 1% glutamine, 1% non-essential amino acids) and frozen immediately on dry ice. We also measured salivation volume from mosquitoes by allowing them to expectorate into 1 µL capillary tubes as described previously and measured the height of the saliva with a digital fractional caliper (presicion: ±0.02 mm). The volume of saliva was calculated using the cylinder volume formula (V = π(r^2^ × h) = 3.1416 × 0.01 × h = mm^3^ or µL). Following expectoration, salivary glands were dissected from the same mosquitoes that provided saliva and placed in 500 μL of DMEM medium (7% of heat inactivated FBS, 1% penicillin/streptomycin, 1% glutamine, 1% non-essential amino acids). The salivary gland samples were frozen immediately (dry ice) prior to determining CHIKV titers. CHIKV titers in saliva and salivary glands were reported as pfu/mL from plaque assays as previously described. 

### 2.5. Statistical Analyses

The proportion of infected salivary glands and saliva were calculated by averaging the number of infected samples divided by the total tested. All data were analyzed with GraphPad prism software (version 5.0, La Jolla, CA, USA) for testing significant differences (*p* < 0.05) in DIR and TER (Fisher’s exact test, χ^2^ test) and the correlation among virus titers in saliva and salivary glands (Pearson correlation). Analysis of variance (one-way and two-way ANOVA) was used to determine the statistical significance and CORREL function (Excel 2016, Microsoft, Redmond, DC, USA) for determining the correlation coefficient. 

## 3. Results

### 3.1. Dissemination of CHIKV R99659 and LR2006 OPY1 to Salivary Glands of Ae. albopictus Mosquito Strains

The DIR of *Ae. albopictus* for CHIKV was evaluated 9 days after infection with R99659 (blood meal virus titer: 5.6 × 10^7^ pfu/mL) or LR2006 OPY1 (blood meal virus titer: 6.6 × 10^7^ pfu/mL). We detected no significant difference in DIR among the *Ae. albopictus* strains (Florida, Louisiana, New Jersey, and La Réunion) infected with either R99659 and LR2006 OPY1 (*p* = 0.3916). DIR ranged from 93% to 96.6% for R99659 and 93.3% to 100% for LR2006 OPY1. However, significant differences were detected in the virus titers of salivary glands among the *Ae. albopictus* strains infected with either R99659 (*p* < 0.0001) or LR2006 OPY1 (*p* = 0.0461; Figure 1). R99659 virus titers in the salivary glands of the La Réunion strain were significantly higher than titers in the New Jersey strain (*p* = 0.0010) or Louisiana strain (*p* = 0.0181). LR2006 OPY1 virus titers in the New Jersey strain were significantly lower than in the Florida and La Réunion strains of *Ae. albopictus* (*p* = 0.0330 and *p* = 0.0282, respectively; Figure 1). While the LR2006 OPY1 virus titers were higher than R99659 virus titers in the salivary glands of all *Ae. albopictus* strains tested, statistically significant differences were detected between R99659 and LR2006 OPY1 virus titers in the La Réunion (*p* = 0.0453), Florida (*p* = 0.0089) and New Jersey (*p* = 0.0472) strains. We observed no significant differences in DIR between the two CHIKV isolates in the Louisiana strain (*p* = 0.1920).

### 3.2. CHIKV R99659 and LR2006 OPY1 Viruses in Saliva of Ae. albopictus Mosquito Strains

We did not detect significant differences in virus prevalence in saliva (TER) among the *Ae. albopictus* strains infected with R99659 virus (*p* = 0.3916) or the *Ae. albopictus* strains infected with LR2006 OPY1 virus. The prevalence (TER) of virus in saliva ranged from 33.3% to 56.6% for R99659 virus and 80% to 86.6% for CHIKV LR2006 OPY1 virus. The TER of R99659-infected *Ae. albopictus* strains was significantly lower than mosquito strains infected with LR2006 OPY1 virus (*p* < 0.0001; Figure 2). Significant differences were also observed for virus titers in saliva among *Ae. albopictus* strains infected with R99659 (*p* = 0.012. Figure 2). No significant differences were observed for LR2006 OPY1 (*p* = 0.7544. Figure 2) titers in saliva of the four *Ae. albopictus* strains. The LR2006 OPY1 titers in saliva of the New Jersey strain were significantly higher than R99659 titers in saliva of the same strain (*p* < 0.0001) possibly indicating LR2006 OPY1 virus has a fitness advantage in these mosquitoes.

### 3.3. Dissemination and Transmission Potential of CHIKV R99659 and LR2006 OPY1 in Ae. aegypti (Poza Rica)

The DIRs of *A. aegypti* (Poza Rica) for R99659 (blood meal virus titer 6.2 × 10^7^ pfu/mL) or LR2006 OPY1 (blood meal virus titer 6.6 × 10^7^ pfu/mL) were also evaluated 9 days post-infection. DIRs were high for all CHIKV isolates tested in *Ae. aegypti*, with no significant difference (*p* = 0.3679) between R99659 and LR2006 OPY1 viruses (Figure 3). The DIRs were 92% for R99659 and 93% for LR2006 OPY1 viruses (Figure 3). The TER for *Ae. aegypti* (Poza Rica) infected with LR2006 OPY1 and R99659 viruses were statistically similar (*p* > 0.05). These data indicated that *Ae. aegypti* (Poza Rica) strains had statistically similar vector competence (DIR × TER) for these viruses. DIR, TER and vector competence for the two CHIKVs and the four *Ae. albopictus* strains and one *Ae. aegypti* strain are summarized in Table 1. 

### 3.4. Relationship between CHIKV Infectious Particles in Saliva and Salivary Glands

We observed no correlation between salivary gland virus titer (pfu/mL) and saliva virus titer (pfu/mL) for R99659 and LR2006 OPY1 viruses in *Ae. aegypti* (Poza Rica; Figure 4A). This was also true for the four *Ae. albopictus* strains infected with R99659 genotype (Figure 4B) and the Florida and Louisiana *Ae. albopictus* strains infected with LR2006 OPY1 (Figure 4C). However, the TER of the La Réunion *Ae. albopictus* strain infected with LR2006 OPY1 showed a positive correlation between salivary gland virus titer and saliva virus titer (correlation coefficient = 0.672). Although the New Jersey *A. albopictus* strain had a positive correlation between the two variables, it was weak and likely insignificant (correlation coefficient = 0.323; Figure 4C).

The presence of a salivary gland escape barrier (SGEB) was determined by dividing the number of saliva samples without virus by the number of virus positive salivary glands. The highest SGEB occurred with the *Ae. aegypti* (Poza Rica) strain since many mosquitoes were without detectable virus in their saliva even though their salivary glands were highly infected and all mosquitoes were able to salivate regardless of their infection status. This was also true for the four *Ae. albopictus* strains tested but the correlation was weaker than the SGEBs associated with *Ae. aegypti*. Interestingly, SGEB was dependent upon the viral genotype, since the four *Ae. albopictus* strains and the *Ae. aegypti* strain infected with R99659 (Asian genotype) showed a higher SGEB than mosquitoes infected with LR2006 OPY1 virus. The New Jersey *Ae. albopictus* strain infected with R99659 virus presented the highest SGEB (64.3 %) compared with the Florida, La Réunion or Ae. aegypti (Poza Rica) strains (SGEB = 44.4%, 44.8%, 48% and 43.5% respectively; Figure 4A–C). In contrast, only 10.7% to 20% of mosquitoes presented a SGEB with LR2006 OPY1 (ECSA IOL genotype), however, the titer of virus in the salivary glands did vary significantly between mosquitoes infected with R99659 and LR2006 OPY1. 

### 3.5. Vector Competence of Ae. aegypti (Poza Rica) and Ae. albopictus Strains for R99659 versus LR2006 OPY1 Viruses

We multiplied DIR by TER to determine vector competence of each group of mosquitoes (*n* = 30/group) for each virus. The four *Ae. albopictus* strains had significantly higher vector competence for LR2006 OPY1 virus versus R99659 virus (*p* = 0.0001; Table 1). *Ae. aegypti* (Poza Rica) also had a significantly higher vector competence for LR2006 OPY1 virus than R99659 virus (*p* = 0.005). Comparisons of the vector competence of *Ae. aegypti* versus *Ae. albopictus* for LR2006 OPY1 were not different (*p* = 0.1881). Comparisons of *Ae. aegypti* (Poza Rica) versus the *Ae. albopictus* strains (La Réunion, Louisiana, and Florida) for R99659 were also not statistically different (*p* > 0.05). The only significant differences when comparing vector competence of mosquito groups for R99659 virus was between the New Jersey strain and the Florida or Louisiana *Ae. albopictus* groups (*p* = 0.0191 and *p* = 0.0083).

### 3.6. Measurement of Salivation Using Ae. aegypti as A Model for Determining Saliva Volume from An Aedes Species

In our assay, saliva is expectorated just after the mosquito proboscis immerses into collection fluid. Mosquitoes such as *Culex. pipiens* salivate an average of 4.7 nL during the blood feeding process [26]. The rate of salivation may vary depending on the species, age, size, physiological state of the mosquito and environmental conditions. We initially determined the average of saliva volume produced during probing by uninfected *Ae. aegypti*. Here we used a capillary tube (0.2 mm inner diameter) prefilled with 1 µL of immersion oil type B. Twenty-six mosquitoes were allowed to salivate into the oil at room temperature for 1 h and then measured the height of saliva in the capillary tube with a digital fractional caliper (presicion: ±0.02 mm). The volume of saliva was then calculated using the cylinder volume formula (V = π(r^2^ × h) = 3.1416× 0.01 × h = mm^3^ or µL; Figure 5A). After creating the frequency distribution (histogram) and “Gaussian” distribution using GraphPad Prism software, we calculated that *Ae. aegypti* mosquitoes salivated an average of 6.82 ± 2.88 nL (Figure 5B). We then determined whether the volume of saliva influenced virus titer by collecting the saliva in calibrated capillary tubes and quantifying the expectorate volume. The titer of CHIKV (pfu/mL) did not strongly correlate with the volume of saliva, since volumes 2–3 nL of saliva contained as much or more virus titer than 5–7 nL volumes of saliva (Figure 5C). Figure 5C is one data set from multiple collections of saliva and determinations of CHIKV titer. We have observed little correlation between virus titer and saliva volume for other arboviruses such as Zika and dengue-2 viruses (*Flavivirus*; unpublished data) from hundreds of *Ae. aegypti* saliva samples [27].

## 4. Discussion

Vector competence, the ability of an arbovirus vector to acquire a virus and successfully transmit the virus to another susceptible host, is determined by genetic traits of vector species and vector populations within species. Vector competence, the efficiency of viral replication and external factors including temperature, the availability of vertebrate hosts, vector population density and predation, determine how successful a vector population will be in transmitting virus in nature [28]. DIR and TER analyses of laboratory-infected *Ae. albopictus* and *Ae. aegypti* populations were performed here to indicate vector competence. We reasoned that the high DIR values ranging from 93% to 100% of infected salivary glands for each *Aedes* species lessened the importance of analyzing midgut infection rates 9 days after per os infection.

In our study, the *Ae. albopictus* strains clearly had uniformly high DIR or CHIKV prevalence in salivary glands after infection with R99659 or LR2006 OPY1 virus. Although the DIRs were similar among the *Ae. albopictus* strains tested, the salivary glands infected with the two CHIKVs varied in virus titers. The mean virus titer of both CHIKVs were highest for the La Réunion strain and lowest for the New Jersey strain indicating that geographic origin of the vector is an important factor. Nevertheless, differences in vector competence of the New Jersey strain for LR2006 OPY1 were not significant when compared to the other *Ae. albopictus* strains and *Ae. aegypti* (Poza Rica) (see Table 1). CHIKV studies to date have demonstrated distinct differences in vector competence of *Ae*. *aegypti* and *Ae*. *albopictus* depending on geographic origin of the mosquitoes and the genotype of CHIKV [28,29,30,31,32,33].

Although the *Ae. albopictus* strains had statistically similar DIRs for both CHIKVs, the TER for CHIKV R99659 virus was significantly lower in all mosquito strains (*Ae. albopictus* and *Ae. aegypti*) than the LR2006 OPY1 virus (*p* = 0.012) suggesting the LR2006 OPY1 virus is more efficiently transmitted than R99659 virus. The New Jersey strain had lower TER values than the other *Ae. albopictus* strains tested suggesting that population was less able to transmit CHIKV. Our results support earlier reports that noted similar DIR and TER outcomes for CHIKVs in *Ae. aegypti* and *Ae. albopictus* populations originating in the Americas [32]. 

We found no correlation between the virus titer in the salivary gland and virus titers in the saliva (Figure 4) and demonstrated in our *Ae. aegypti* model that the volume of saliva does not influence virus titers in saliva (Figure 5). Although we have not measured saliva volume from *Ae. albopictus* or correlated their saliva volume with virus titer, this will be a topic of future experiments. CHIKV provides evidence of how an arbovirus can acquire the capacity for efficient transmission by a new vector mosquito species with unfortunate medical outcomes. During the CHIKV’s emergence in the Indian Ocean basin, it acquired a mutation in the coding sequence of the envelope glycoprotein E1 that resulted in the substitution of a valine for an alanine at position 226 (A226V) of E1 [12]. CHIKV with this mutation increased the vector competence of *Ae. albopictus* for LR2006 OPY1 [10]. Interestingly, the CHIKV that was introduced into the Caribbean in 2013 lacked the A226V mutation, and this mutation has not yet been detected in CHIKVs from the Americas [34]. Given that arboviruses are subject to antiviral pressures from the mosquito, arboviral genome diversity in infected mosquito vectors likely allowed selection of CHIKV-IOL that were better adapted to *Ae. albopictus* to ensure the virus’ survival and transmission [35].

The DIR and TER data suggests that the salivary glands of *Aedes* vectors may lead to barriers to CHIKV transmission that vary with mosquito and viral genetics. Our data demonstrated that 93–100% of mosquitoes had salivary gland infections but only 33–86% of saliva samples were positive for the two CHIKVs. This difference was more pronounced in *Ae. aegypti* than *Ae. albopictus*. We also measured saliva volume from *Ae. aegypti* and saw little correlation between the volume of expectorate and virus titer. SGEBs have been reported for La Crosse (LACV) and Sindbis virus transmission by *Aedes* and *Culex* species, respectively [36,37,38,39]. For example, *Aedes hendersoni* was shown to be an incompetent vector of LACV due to a SGEB in which the salivary glands were infected but the mosquito failed to transmit the virus orally [39]. More recently, a SGEB has been reported that affects Rift Valley Fever virus transmission [40]. Molecular mechanisms that explain SGEB for arboviruses will probably include a combination of vector and viral genetic factors that determine the efficiency of arbovirus transmission from a vector population. Arbovirus infection of salivary glands typically begins in the distal lateral lobes [41,42,43]. Certain arboviruses such as CHIKV infect the proximal lateral and median lobes of *Ae. aegypti* [42] and following replication, virus deposits in the apical cavities of acinar cells, which can lead to inoculation of a susceptible host upon refeeding [44]. We demonstrated here that TER in *Ae. aegypti* and *Ae. albopictus* vary according to the origins of mosquito populations tested and the CHIKV strain.

CHIKV is now widespread worldwide and likely will continue to pose a public health threat globally wherever the invasive mosquitoes *Ae. albopictus* and *Ae. aegypti* exists to transmit CHIKV to a susceptible human population. The emergence of mosquito-borne CHIKV in the Americas starting in 2013 led to a geographically widespread outbreak with human illness and the establishment of CHIKV transmission cycles in urban settings. As evidence of this, 11 cases from autochthonous transmission of CHIK were detected in south Florida in 2014 [45] emphasizing the importance of knowing the relative vector competence of North American mosquitoes to this arbovirus. This knowledge, coupled with enhanced understanding of critical arbovirus-vector interactions and the ecology of the major vectors will enable locally specific, efficient deployment of public health resources in the event of another outbreak of CHIKV in North America.

## 5. Conclusions

(1)We infected the *Ae. albopictus* strains and *Ae. aegypti* with blood meals containing 5–7 × 10^7^ PFU/mL of R99659 (Asian genotype) and LR2006 OPY1 (ECSA-IOL genotype) and analyzed the mosquitoes nine days later to detect maximum infection of their salivary glands and saliva.(2)All four *Ae. albopictus* strains were highly susceptible to LR2006 OPY1 and R99659 viruses and their CHIKV disseminated infection rates (DIR) were statistically similar (*p* = 0.3916).(3)We demonstrated here that the TER in *Ae. aegypti* and *Ae. albopictus* vary according to their geographic origins of the mosquitoes and the CHIKV strain contributing to a salivary gland barrier for releasing R99659 virus into saliva.(4)Virus titers in salivary glands did not correlate with saliva volume, but *Ae. aegypti* expectorated on average a volume of 6.8 nL/mosquito.(5)If introduced, LR2006 OPY1 virus could increase the risk of CHIKV transmission by both *Aedes* species in the western hemisphere.

## Figures and Tables

**Figure 1 insects-10-00039-f001:**
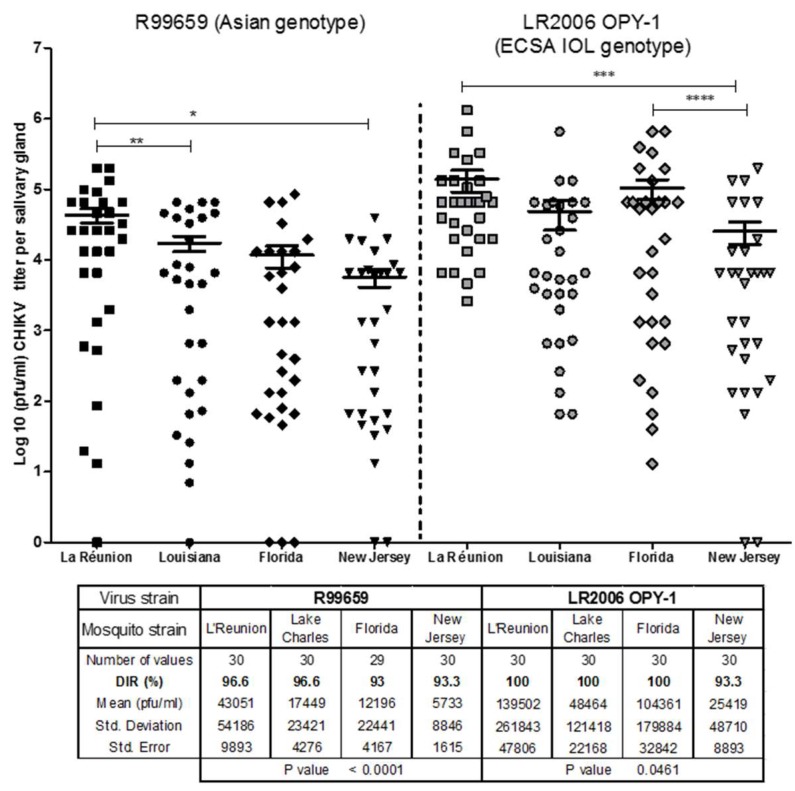
Chikungunya virus (CHIKV) disseminated infection rates (DIR) and titer in salivary glands of four *Ae. albopictus* strains. *Aedes albopictus* were orally infected with either CHIKV R99659 (blood titer 5.6 × 10^7^ pfu/mL) or CHIKV LR2006 OPY1 (blood titer 6.6 × 10^7^ pfu/mL) and the DIR (prevalence) and virus titers were determined 9 days post infection in salivary glands by plaque assay. The number of mosquitoes assayed from each mosquito and CHIKV strain and the prevalence (% infection) are listed. Horizontal bars show mean titer values with standard error of the mean (SEM). *p*-values of virus titers among the *Ae. albopictus* strains infected with either R99659 or LR2006 OPY1 are shown in table. * *p* = 0.0010, ** *p* = 0.0181, *** *p* = 0.0330, **** *p* = 0.0282.

**Figure 2 insects-10-00039-f002:**
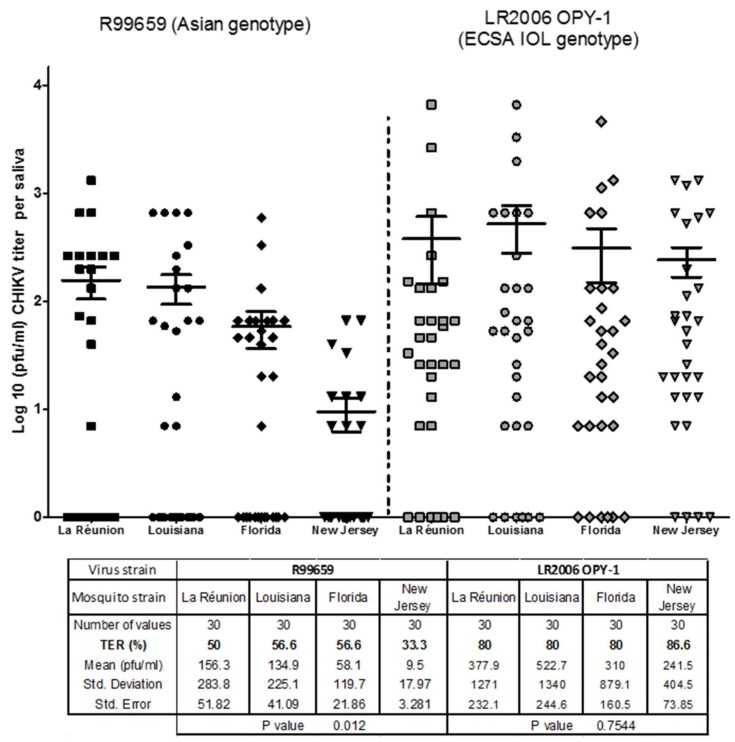
Transmission efficiency rate (TER) assay of CHIKVs in four *Ae. albopictus* strains. The TER (prevalence) and virus titer were determined for *Ae. albopictus* (three U.S. strains and the La Réunion strain) orally infected with either CHIKV R99659 (blood meal virus titer 5.6 × 10^7^ pfu/mL) or CHIKV LR2006 OPY1 (blood meal virus titer 6.6 × 10^7^ pfu/mL). Virus titers were determined 9 days later in saliva by plaque assay. Saliva for each female was collected using forced saliva capillary method. The number of mosquitoes assayed from each mosquito and CHIKV strain and the prevalence (% infection) are listed. Horizontal bars show mean titer values with SEM. *p* value of virus titers among the *Ae. albopictus* strains infected with either R99659 or LR2006 OPY1 were *p* = 0.012 and *p* = 0.7544, respectively.

**Figure 3 insects-10-00039-f003:**
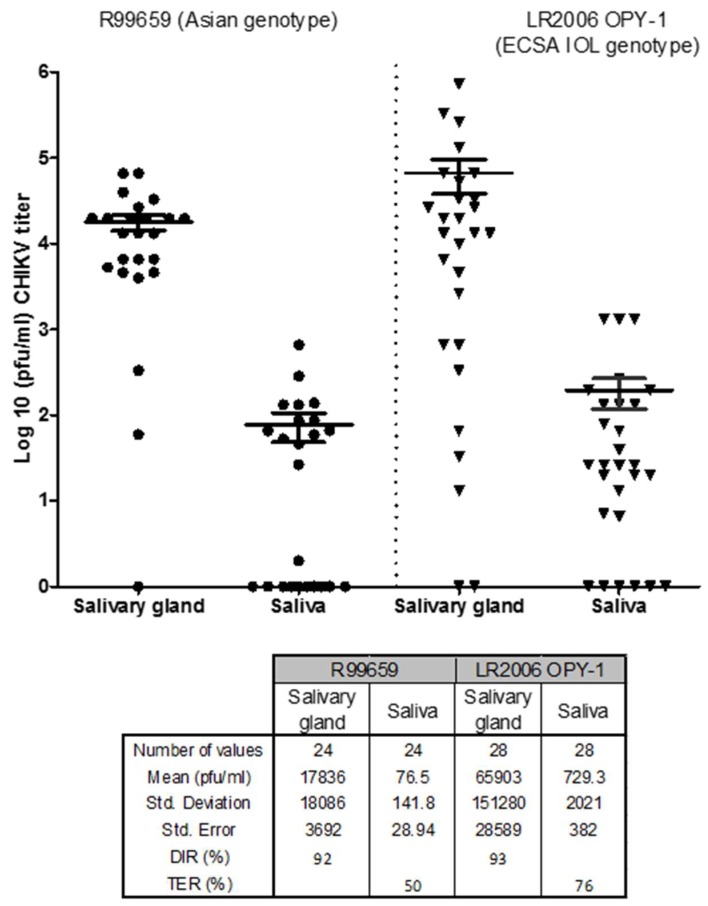
CHIKV titers in Poza Rica *A. aegypti* strain from Mexico. *Ae. aegypti (Poza Rica)* were orally infected with either CHIKV R99659 (blood meal virus titer 6 × 10^7^ pfu/mL) and CHIKV LR2006 OPY1 (blood meal virus titer 3 × 10^7^ pfu/mL). Virus titers were determined 9 days later by plaque assay in salivary glands and saliva. The number of mosquitoes assayed for each CHIKV strain and the prevalence (% infection) of CHIKV in saliva are listed. Horizontal bars show mean titer values with SEM.

**Figure 4 insects-10-00039-f004:**
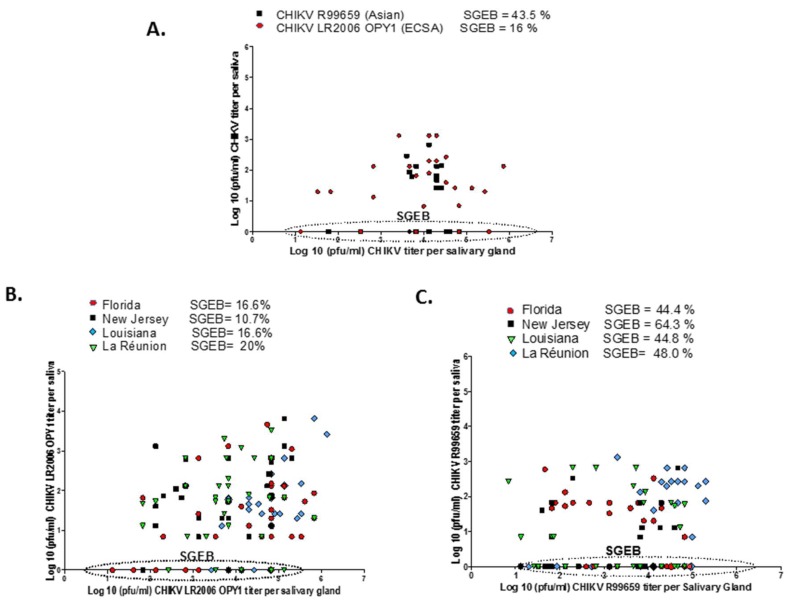
Correlation between CHIKV titers of saliva and salivary glands. The presence of a salivary gland escape barrier (SGEB) was tested by correlating CHIKV titers in salivary glands with virus titers in saliva. Saliva was collected at 9 days post infection and then salivary glands were dissected from the same mosquitoes. Virus titers from salivary glands and saliva were determined by plaque assays. (**A**) Correlation between saliva and salivary gland of *Ae. aegypti* (Poza Rica) infected with CHIKV R99659 (●) and CHIKV LR2006 OPY1 (■). (**B**) Correlation between saliva and salivary gland of *Ae. albopictus* infected with CHIKV R99659. Florida strain (●), New Jersey strain (■), Louisiana, (▼) and La Réunion (♦). (**C**) Correlation between saliva and salivary gland of *Ae. albopictus* infected with CHIKV LR2006 OPY1. Florida strain (●), New Jersey strain (■), Louisiana, (▼) and La Réunion (♦). The % SGEB is the number of saliva samples without virus divided by the number of virus positive salivary gland samples. Percentage of mosquitoes with SGEB are listed for each group.

**Figure 5 insects-10-00039-f005:**
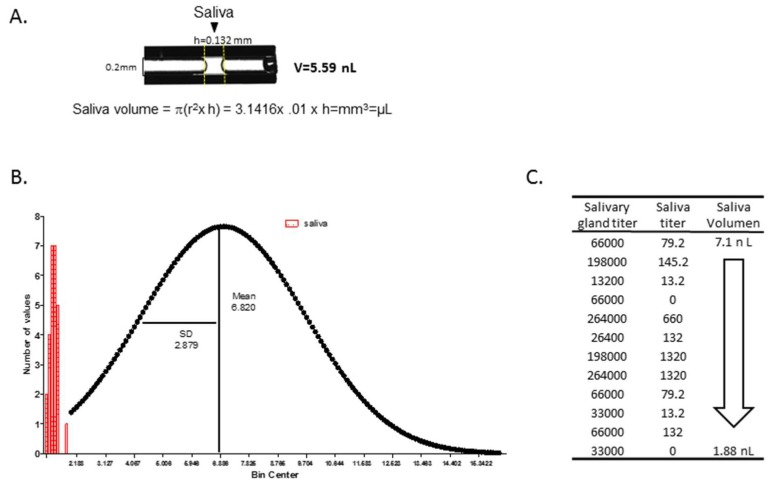
Frequency and Gaussian distribution of saliva volume in *Ae. aegypti*. The average saliva volume produced by *Ae. aegypti*, was determined by forcing mosquitoes to salivate into a calibrated capillary tube (0.2 mm inner diameter) filled with immersion oil type B. Twenty six mosquitoes were allowed to salivate into the oil at room temperature for 1 h. (**A**) An image of a capillary with saliva was captured after the mosquito expectorated. The height of saliva was measured using a digital fractional caliper (presicion: ±0.02 mm). The volume of saliva was calculated using the cylinder volume formula (V = π(r^2^ × h) = 3.1416× 0.01 × h = mm^3^ or µL). (**B**) Frequency distribution (histogram) and Gaussian distribution using GraphPad Prism software. The mean value at the center of the distribution and SD is a measure of the width of the distribution. *Aedes* mosquitoes salivated an average of 6.82 ± 2.88 nL. (**C**) R99659 titers were determined from a range of saliva volumes between 1.8 and 7 nL. We did not see a strong correlation between virus titers in salivary gland and the volume of saliva. Arrow indicates saliva descending volumes between 7.1 nL and 1.88 nL.

**Table 1 insects-10-00039-t001:** DIR, TER, and vector competence for R99659 and LR2006 OPY1 CHIKVs and the four *Ae. albopictus* strains and one *Ae. aegypti* strain. *n* = 30 for all pairwise test.

**R99659 CHIKV R99659**			
***Ae. albopictus***	**Dissemination Rates (DIR)**	**Transmission Efficiency Rate (TER)**	**Vector Competence (VC)**
La Réunion	96.6	50	48.3
Louisiana	96.6	56.6	54.7
Florida	93	56.6	52.7
New Jersey	93.3	33.3	31.1
**LR2006 OPY1 CHIKVOPY1 ZIKVOPY-1**			
***Ae. albopictus***	**Dissemination Rates (DIR)**	**Transmission Efficiency Rate (TER)**	**Vector Competence (VC)**
La Réunion	100	80	80
Louisiana	100	80	80
Florida	100	80	80
New Jersey	93.3	86.6	80.1
**Aedes aegypti (Poza Rica)**			
**ZIKV**	**Dissemination Rates (DIR)**	**Transmission Efficiency Rate (TER)**	**Vector Competence (VC)**
R99659	92	50	46
LR2006 OPY1	93	76	70.6

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
