# Peer review of "Analysis of Salivary Glands and Saliva from Aedes albopictus and Aedes aegypti Infected with Chikungunya Viruses"

_insects, 2019, doi:10.3390/insects10020039_

Round 1
Reviewer 1 Report
Review of Sanchez-Vargas et al Submitted to Insects
Reviewer Summary:
Sanchez Vargas et al sought to determine the transmission potential for two isolates of CHIKV to three populations of Aedes albopictus mosquitoes from the U.S. and one from La Reunion Island, as well as a population of Aedes aegypti mosquitoes collected from Mexico. One of the CHIKV isolates, LR2006, emerged during an outbreak of CHIKV on La Reunion Island in 2006, and exhibited an enhanced ability to infect and subsequently be transmitted by Aedes albopictus mosquitoes. The authors examined salivary gland infection by the CHIKV viruses as well as secreted virus in the saliva to measure the vector competence of these mosquito populations, and thus their potential as vectors for CHIKV in the U.S. The authors describe several important findings related to CHIKV transmission dynamics in relation to these mosquito populations:
· All Aedes albopictus and Aedes aegypti mosquitoes exhibited similar rates of salivary gland infection (DIR) to both CHIKV R99659 and LR2006 viruses, showing that this virus readily disseminates from the midgut of these mosquitoes and largely excluding the presence of a significant midgut escape barrier to these viruses, at least under experimental conditions.
· Ae. albopictus mosquitoes from New Jersey more weakly supported replication by either virus in the salivary glands compared to the other three populations or that of Aedes aegypti.
· Despite nearly identical rates of salivary gland infection, the transmission rate of CHIKV LR2006 (as measured by presence of the virus in mosquito saliva) greatly exceeded that of the R99659 virus, implicating that a virus strain-specific salivary gland escape barrier exists in these mosquitoes. Therefore, the vector competence of all of the mosquito populations was significantly higher for the LR2006 virus than the R99659 virus. This observation was also made for Ae. aegypti mosquitoes.
· In general, there was not a strong relationship between virus titer in the salivary gland and virus titer in the saliva, though some degree of correlation did exist in the La Reunion Island and New Jersey populations of Ae. albopictus.
· There was not a high degree of correlation between the volume of saliva and the titer of virus in the saliva.
I found the manuscript to be well written, and the methods and conclusions to be scientifically sound. I find this manuscript appropriate to be published in Insects. I have only the following minor comments:
· There appear to be font style/color shifts at several points throughout the manuscript. I am not sure whether this was in the original submitted manuscript or is an artifact of the formatting done at the journal prior to sending out for review.
· The Statistical Analyses section of the Materials and Methods is not sufficient in my opinion. For example, it states that ANOVA was used, but does not indicate what post-tests (i.e. multiple-comparisons) were used, and if multiplicity-adjusted p-values are what are being reported.
· Statistically significant comparisons between mosquito populations is not noted on the figures themselves. Though correctly noted in the text, and in general relatively easy to determine by eye, it would aid if visual indication of statistically significant comparisons were made on the figures themselves.
· It is not obvious in Fig 2 what the p-values at the bottom of the chart are indicating, as they are both indicating different measurements. Below the R99659 header, the p-value appears to indicate the difference in virus titers between mosquitoes, while under the LR2006 header, the p-value indicates the non-significant difference in the prevalence (TER) of virus in the saliva. While this is indicated (albeit, somewhat confusingly) in the text, it could be more clearly defined in the figure itself. Related to this, the use of the term TER in lines 171-173 and 176-179 is confusing, because one is in relation to the TER itself (prevalence), and the other is in relation to the titer. I would suggest removing the term TER from lines 176-179 to avoid any confusion, as at first glance the two sentences appear to contradict one another (“…no significant differences in TER” vs. “significant differences in TER…”).
· Table 1 could benefit from Headers above the tables themselves, as opposed to relying on the order given in the caption to determine which table is which.
· Line 275: Are these published or unpublished observations? If published, a citation is warranted, if not, a note that they are unpublished observations should be made.
· Line 338: “…have been reported fhat” should be “…that”
· Lines 305-311: It is stated here that the NJ population of Ae. albopictus is likely to be a less competent vector of CHIKV, and I agree that is possible, since a lower titer of virus in saliva would presumably (though not definitively) decrease the likelihood of transmission to/infection of the vertebrate host. However, with regards to how the authors actually measure vector competence, the NJ population is equally as competent as the other Ae. albopictus populations and the Ae. aegypti population (as detailed in Table 1), at least for the LR2006 virus.
Author Response
Response
Reviewer 1
Comments and Suggestions for Authors
The manuscript by Vargus presents some quite interesting data on vector competence of Ae. albopitus (and aegypti) for chikungunya viruses. These data enhance our understanding of the interaction of mosquito and viral genetics that ultimately are important in virus transmission cycles. The studies reported here appeared to be competently executed and comprehensive with regard to the questions being asked, although only a single strain of Ae. aegypti was used as a comparator. Yes, only Aedes aegypti (Poza Rica) was used as a comparator for the Ae. albopictus strains and as model for saliva collection therefore the species was included in the title. We are currently conducting CHIKV experiments with a number of different Ae. aegypti populations.
I do not have major criticisms, but offer the following suggestions:
The authors assert differences in virus replication dependent on geographic origin of the vector. Is “geographic origin” actually the best term to use in this context? All of the mosquito strains were obtained by CDC, but there is not a description of how long they had been colonized, which might have an effect on their competence for CHIKV.
We retain the phrase “geographic origin” but state that mosquito strains were in colony for less than 10 generations for New Jersey, La Réunion and Poza Rica strains but Florida and Louisiana strain colonization are most likely a greater number of generations.
L19-20: saliva can be infectious and contain virus but cannot be infected.
Good point. This has been changed where it appears in the original text.
L21-23: Clumsy sentence – perhaps modify as “The transmission efficiency rate (TER) was significantly lower for R99659 virus compared to LR2006 OPY1 virus in all Ae. albopictus strains and Ae. aegypti (Poza Rica) (P=0.012).”
Changed in the abstract to reflect suggestion.
L46: delete “based on their RNA genome sequence” (what else would genotypes be based on?). You are right; deleted.
L60: Suggest changing to “Our goal was to analyze salivary gland infection and the titer of virus in saliva”. Changed to reflect suggestion.
Line 88-9: Suggest changing to “and maintained with 5% CO2 at 37ºC and 28ºC, respectively. Changed to reflect suggestion.
Throughout: check capitalization e.g. glutamine, non-essential amino acids.
Changed to lower case.
L140:delete “P < 0.05” (stated at the end of the paragraph).
Deleted.
Figures 1 and 2: In the table “Prevalence” is a strange term – why not just DIV or TER? Mean is virus titer (pfu/ml) – you have plenty of room to expand laterally and that would enhance clarity. Also, I would suggest exchanging the top (virus strain) and second (mosquito) rows of both tables.
Prevalence has been changed to DIR and TER in figures. Adjustments have been made to both tables for clarity.
Table 1 needs some subheadings to indicate which group of rows is which virus (you can determine this, but it would be useful for clarity. You might also indicate in the caption that these data represent all viruses (for the mosquito strains) and all mosquito strains (for the two viruses).
Subheadings have been added to the tables for clarification.
Figure 3: is “bloodfed” does not appear useful as a row in the table?
Agree, Deleted.
L260-268: Considerable redundancy with description in Methods section.
We agree, but we think it is worthwhile to restate for readers of this result section.
Line 275: insert “unpublished data” ?
Added
Figure 5: It seems quite strange that 11 of the 12 volumes were exactly 7.1 nl – is that correct??
The arrow depicts descending saliva volumes between 7.1 nL and 1.88 nL. Due to the number of samples the displayed data set “volumes” were rapidly screened by height of saliva in the capillary tube and did not do formal volume calculations. Nevertheless we are certain the data sets show volumes that would fall in volumes between the two extremes.
Reviewer 2 Report
The manuscript by Vargus presents some quite interesting data on vector competence of Ae. albopitus (and aegypti) for chikungunya viruses. These data enhance our understanding of the interaction of mosquito and viral genetics that ultimately are important in virus transmission cycles. The studies reported here appeared to be competently executed and comprehensive with regard to the questions being asked, although only a single strain of Ae. aegypti was used as a comparator. I do not have major criticisms, but offer the following suggestions:
The authors assert differences in virus replication dependent on geographic origin of the vector. Is “geographic origin” actually the best term to use in this context? All of the mosquito strains were obtained by CDC, but there is not a description of how long they had been colonized, which might have an effect on their competence for CHIKV.
L19-20: saliva can be infectious and contain virus but cannot be infected
L21-23: Clumsy sentence – perhaps modifyas “The transmission efficiency rate (TER) was significantly lower for R99659 virus compared to LR2006 OPY1 virus in all Ae. albopictus strains and Ae. aegypti (Poza Rica) (P=0.012).”
L46: delete “based on their RNA genome sequence” (what else would genotypes be based on?)
L60: Suggest changing to “Our goal was to analyze salivary gland infection and the titer of virus in saliva”
Line 88-9: Suggest changing to “and maintained with 5% CO2 at 37ºC and 28ºC, respectively.
Throughout: check capitalization e.g. glutamine, non-essential amino acids
L140:delete “P < 0.05” (stated at the end of the paragraph)
Figures 1 and 2: In the table “Prevalence” is a strange term – why not just DIV or TER? Mean is virus titer (pfu/ml) – you have plenty of room to expand laterally and that would enhance clarity. Also, I would suggest exchanging the top (virus strain) and second (mosquito) rows of both tables.
Table 1 needs some subheadings to indicate which group of rows is which virus (you can determine this, but it would be useful for clarity. You might also indicate in the caption that these data represent all viruses (for the mosquito strains) and all mosquito strains (for the two viruses).
Figure 3: is “bloodfed” does not appear useful as a row in the table?
L260-268: Considerable redundancy with description in Methods section.
Line 275: insert “unpublished data” ?
Figure 5: It seems quite strange that 11 of the 12 volumes were exactly 7.1 nl – is that correct??
Author Response
Responses
Reviewer 2.
Sanchez Vargas et al sought to determine the transmission potential for two isolates of CHIKV to three populations of Aedes albopictus mosquitoes from the U.S. and one from La Reunion Island, as well as a population of Aedes aegypti mosquitoes collected from Mexico. One of the CHIKV isolates, LR2006, emerged during an outbreak of CHIKV on La Reunion Island in 2006, and exhibited an enhanced ability to infect and subsequently be transmitted by Aedes albopictus mosquitoes. The authors examined salivary gland infection by the CHIKV viruses as well as secreted virus in the saliva to measure the vector competence of these mosquito populations, and thus their potential as vectors for CHIKV in the U.S. The authors describe several important findings related to CHIKV transmission dynamics in relation to these mosquito populations:
· All Aedes albopictus and Aedes aegypti mosquitoes exhibited similar rates of salivary gland infection (DIR) to both CHIKV R99659 and LR2006 viruses, showing that this virus readily disseminates from the midgut of these mosquitoes and largely excluding the presence of a significant midgut escape barrier to these viruses, at least under experimental conditions.
· Ae. albopictus mosquitoes from New Jersey more weakly supported replication by either virus in the salivary glands compared to the other three populations or that of Aedes aegypti.
· Despite nearly identical rates of salivary gland infection, the transmission rate of CHIKV LR2006 (as measured by presence of the virus in mosquito saliva) greatly exceeded that of the R99659 virus, implicating that a virus strain-specific salivary gland escape barrier exists in these mosquitoes. Therefore, the vector competence of all of the mosquito populations was significantly higher for the LR2006 virus than the R99659 virus. This observation was also made for Ae. aegypti mosquitoes.
· In general, there was not a strong relationship between virus titer in the salivary gland and virus titer in the saliva, though some degree of correlation did exist in the La Reunion Island and New Jersey populations of Ae. albopictus.
· There was not a high degree of correlation between the volume of saliva and the titer of virus in the saliva.
I found the manuscript to be well written, and the methods and conclusions to be scientifically sound. I find this manuscript appropriate to be published in Insects. I have only the following minor comments:
There appear to be font style/color shifts at several points throughout the manuscript. I am not sure whether this was in the original submitted manuscript or is an artifact of the formatting done at the journal prior to sending out for review.
This has been corrected.
· The Statistical Analyses section of the Materials and Methods is not sufficient in my opinion. For example, it states that ANOVA was used, but does not indicate what post-tests (i.e. multiple-comparisons) were used, and if multiplicity-adjusted p-values are what are being reported.
The statistical analysis section has been modified for clarity.
· Statistically significant comparisons between mosquito populations is not noted on the figures themselves. Though correctly noted in the text, and in general relatively easy to determine by eye, it would aid if visual indication of statistically significant comparisons were made on the figures themselves.
The statistical comparisons have been stated more clearly in the figure legends.
· It is not obvious in Fig 2 what the p-values at the bottom of the chart are indicating, as they are both indicating different measurements. Below the R99659 header, the p-value appears to indicate the difference in virus titers between mosquitoes, while under the LR2006 header, the p-value indicates the non-significant difference in the prevalence (TER) of virus in the saliva. While this is indicated (albeit, somewhat confusingly) in the text, it could be more clearly defined in the figure itself. Related to this, the use of the term TER in lines 171-173 and 176-179 is confusing, because one is in relation to the TER itself (prevalence), and the other is in relation to the titer. I would suggest removing the term TER from lines 176-179 to avoid any confusion, as at first glance the two sentences appear to contradict one another (“…no significant differences in TER” vs. “significant differences in TER…”).
This section has been reworded for clarity.
· Table 1 could benefit from Headers above the tables themselves, as opposed to relying on the order given in the caption to determine which table is which.
Corrected. See response to reviewer 1.
· Line 275: Are these published or unpublished observations? If published, a citation is warranted, if not, a note that they are unpublished observations should be made.
Unpublished observations; added.
· Line 338: “…have been reported fhat” should be “…that”.
Corrected.
· Lines 305-311: It is stated here that the NJ population of Ae. albopictus is likely to be a less competent vector of CHIKV, and I agree that is possible, since a lower titer of virus in saliva would presumably (though not definitively) decrease the likelihood of transmission to/infection of the vertebrate host. However, with regards to how the authors actually measure vector competence, the NJ population is equally as competent as the other Ae. albopictus populations and the Ae. aegypti population (as detailed in Table 1), at least for the LR2006 virus.
This has been addressed lines 313-314.